# Effects of Macrofaunal Recolonization on Biogeochemical Processes and Microbiota—A Mesocosm Study

Annika Fiskal [1,*,†,‡], Aixala Gaillard [1,‡], Sebastien Giroud [1,‡], Dejan Malcic [1], Prachi Joshi [1,§], Michael Sander [1], Carsten J. Schubert [1,2] and Mark Alexander Lever [1,*,†]

1    Institute of Biogeochemistry and Pollutant Dynamics (IBP), ETH Zurich, Universitätstrasse 16, 8092 Zurich, Switzerland; gaixala@student.ethz.ch (A.G.); Sebastien.Giroud@eawag.ch (S.G.); dmalcic@student.ethz.ch (D.M.); prachi.joshi@uni-tuebingen.de (P.J.); michael.sander@env.ethz.ch (M.S.); carsten.schubert@eawag.ch (C.J.S.)

2    Department of Surface Waters—Research and Management, Swiss Federal Institute of Aquatic Science and Technology (EAWAG), Seestrasse 79, 6047 Kastanienbaum, Switzerland

*    Correspondence: annikafiskal@gmail.com (A.F.); mark.lever@usys.ethz.ch (M.A.L.)

†    Current address: Department U2-Microbial Ecology, German Federal Institute of Hydrology, Am Mainzer Tor 1, 56068 Koblenz, Germany.

‡    Co-first authors, contributed equally to the study.

§    Current address: Geomicrobiology, University of Tübingen, Schnarrenbergstraße 94-96, 72076 Tübingen, Germany.

**Abstract:** Macroinvertebrates are widespread in lake sediments and alter sedimentary properties through their activity (bioturbation). Understanding the interactions between bioturbation and sediment properties is important given that lakes are important sinks and sources of carbon and nutrients. We studied the biogeochemical impact of macrofauna on surface sediments in 3-month-long mesocosm experiments conducted using sediment cores from a hypoxic, macrofauna-free lake basin. Experimental units consisted of hypoxic controls, oxic treatments, and oxic treatments that were experimentally colonized with chironomid larvae or tubificid worms. Overall, the presence of $O_2$ in bottom water had the strongest geochemical effect and led to oxidation of sediments down to 2 cm depth. Relative to macrofauna-free oxic treatments, chironomid larvae increased sediment pore water concentrations of nitrate and sulfate and lowered porewater concentrations of reduced metals ($Fe^{2+}$, $Mn^{2+}$), presumably by burrow ventilation, whereas tubificid worms increased the redox potential, possibly through sediment reworking. Microbial communities were very similar across oxic treatments; however, the fractions of *α-*, *β-*, and *γ-Proteobacteria* and *Sphingobacteriia* increased, whereas those of *Actinobacteria*, *Planctomycetes*, and *Omnitrophica* decreased compared to hypoxic controls. Sediment microbial communities were, moreover, distinct from those of macrofaunal tubes or feces. We suggest that, under the conditions studied, bottom water oxygenation has a stronger biogeochemical impact on lacustrine surface sediments than macrofaunal bioturbation.

**Keywords:** bioturbation; ventilation; bioirrigation; reworking; microbial community; sediment biogeochemistry; temperate lake sediment; eutrophication; oligotrophication

## 1. Introduction

Bioturbation is defined as the translocation of particles (reworking) and movement of water (ventilation) through sediments by living organisms [1]. Ventilation causes the advective and diffusive exchange of solutes between sediments and overlying water (bioirrigation). This leads to input of oxygen ($O_2$) into otherwise anoxic sediment layers and can cause fluctuations between anoxic/oxic conditions within sediments [1,2]. Reworking due to macrofaunal burrowing, feeding, and defecation activities moves sediment particles in an undirected or directed manner and can thereby mix, oxidize, and introduce organic matter to surface sediments [1,2]. The dominant mode of bioturbation, and thus the

biogeochemical and sedimentological impact of bioturbation on sediments, varies between different macrofaunal species and habitats [1].

In addition bioturbation affects biogeochemical processes by influencing microbial activity [3]. Mechanical breakdown of large into small, more accessible organic particles during macrofaunal feeding promotes microbial colonization and organic matter mineralization [4]. Pumping of high-energy electron acceptors, e.g., $O_2$, nitrate ($NO_3^-$), by ventilation into deeper layers, where these electron acceptors are otherwise absent, interferes with the vertical separation of microbial respiration reactions based on energy yields [5] and causes coexistence of otherwise competing reactions [6,7]. Ventilation can also stimulate microbial activity by removing toxic metabolites, such as $H_2S$ [8], or by providing food to microorganisms ("gardening") [9]. In addition, macrofaunal reworking can decrease microbial populations, e.g., through grazing and deposit-feeding [1]. While most studies have focused on the impact of bioturbation in marine sediments, the effects of bioturbation on freshwater sediments are less studied and understood. The main bioturbators of freshwater sediments include fish, clams, insect larvae and oligochaetes [10,11].

Particularly chironomid larvae and oligochaetes are important and abundant bioturbators across many freshwater sediments [12]. Many chironomid larvae build sedimentary tubes from salivary silk and ventilate these with overlying water, thereby causing bioirrigation of the surrounding sediment [13,14]. Ventilation by chironomid larvae also creates redox interfaces and affects nutrient cycling [1,15] by increasing fluxes of phosphate ($PO_4^{3-}$) and ammonium ($NH_4^+$) to bottom water [16,17], or by promoting phosphorus removal by adsorption [18]. Input of $O_2$ by ventilation may also stimulate nitrification and subsequent denitrification in surrounding sediment [19,20].

The majority of freshwater oligochaetes fall into the family *Tubificidae*. Many members of this family build dense burrow networks into deeper parts of the sediment [21] and feed head-down several centimeters below the surface and defecate at the sediment surface [12]. This activity transports reduced sediment to the surface, where the sediment rapidly oxidizes. The subsequent downward transport of oxidized sediment through the addition of new fecal layers increases the redox potential in surface sediments [22]. Though tubificids typically affect nitrogen (N) and P cycling less than chironomids [20,23], they can also increase the efflux of phosphate and ammonium to overlying water [23,24] or influence rates of nitrification and denitrification [20].

Increased anthropogenic nutrient inputs, especially P, stimulate water column primary production and thereby increase organic carbon (OC) loading and OC burial in lakes through a process called eutrophication [25–27]. Increased OC loading increases $O_2$ consumption and can lead to water column hypoxia or even anoxia, and thereby severely impact ecosystem functioning [27]. Periods of hypoxia or anoxia for weeks to months kill macroinvertebrates or force these to retreat to oxic parts of lakes [28,29]. Decreased bottom water $O_2$ concentrations furthermore alter macrofaunal communities due to differences in physiological tolerance to hypoxic or anoxic conditions among macrofaunal species [30]. Thus, macrofaunal assemblages are studied as indicators of eutrophication [10,31]. To prevent anoxia, the decrease of P input and artificial aeration of water columns have been implemented and led to the macrofaunal recolonization of sediment in eutrophic lakes [27].

So far, little is known about the impact of sediment macrofauna on microbial community structure in lake sediments, even though the observed effects of macrofauna on biogeochemical processes are largely mediated by microorganisms. Among the limited number of studies, several have found that microbial population size does not change or increases slightly due to lacustrine sediment bioturbation [32,33], while others have suggested that predation by oligochaetes may reduce microbial population size [21]. Further studies have shown that bioturbation locally changes sediment bacterial community structure [34,35], e.g., by leading to elevated abundances of methane- and iron (Fe)-oxidizing bacteria in tubes of chironomid larvae [9].

In this study, we investigate experimentally how the switch from severe bottom water hypoxia to oxic conditions, as occurring during the reversal of eutrophication (oligotroph-

ication) or due to artificial aeration of eutrophic lakes, and subsequent colonization by chironomid larvae and tubificid worms, affects sediment biogeochemistry and microbial community structure. To this end, we artificially oxygenated and macrofaunally colonized sediment cores from the hypoxic, macrofauna-free deep basin of Lake Zurich (Switzerland) in the laboratory. Over a period of 3 months, we compare the response of redox-sensitive biogeochemical processes and microbial community composition across hypoxic controls (C), oxic treatments (O), oxic treatments supplemented with chironomid larvae (L), and oxic treatments supplemented with tubificid oligochaete worms (W). We study the impact of oxygenation and bioturbation based on (1) biogeochemical properties of solute (anions, cations) and solid phases (acid extractable Fe, Chl *a*, luminophore beads), (2) microprofiling of $O_2$, redox potential, pH, and $H_2S$, and (3) microbial community analyses based on 16S rRNA gene copy numbers and sequence compositions. Our results provide novel insights into how bottom water oxygenation and recolonization by macrofauna affect sedimentary C, N, P and Fe cycling and microbial community composition in lake sediments.

## 2. Materials and Methods

### 2.1. Core and Macrofaunal Sampling

We obtained 24 sediment cores from the deep basin of Lake Zurich at 137 m water depth (47°16.995 N, 8°35.624 E). The bottom water at the station is hypoxic and sediments are devoid of macrofauna [36]. Cores were on average ~50 cm long and were obtained by gravity coring using 60 cm long and 15 cm wide core liners (UWITEC, Mondsee, Austria). Immediately after retrieval, cores were capped and cooled in ice water during transport to the laboratory at Eawag (Kastanienbaum, Switzerland), where cores were stored at 10 °C until the start of the experiments. An additional 10 sediment cores were obtained from both the shallow sublittoral zone of Lake Lucerne at 24 m water depth (47°00.051 N, 8°20.218 E) and from the deepest part of Lake Baldegg at 66 m water depth (47°11.929 N, 8°15.613 E). These two stations were chosen because of previously described macrofaunal abundance and community structure. Macrofauna at the station in Lake Lucerne almost exclusively consist of chironomid larvae (dominant groups: *Procladius* sp., *Micropsectra* sp., *Macropelopia fehlmanni*, *Tanytarsus* sp., *Sergentia coracina*), whereas the station in Lake Baldegg is dominated by oligochaete worms (dominant groups: unclassified *Tubificidae* without bristles, *Limnodrilus hoffmeisteri*, *L. profundicula*) [37]. Cores with macrofauna were also cooled with ice water during transport and stored at 10 °C at the laboratory, however, overlying water was bubbled with air to prevent hypoxia. Macrofauna were sampled by sieving through 200 μm mesh.

### 2.2. Experimental Design

Cores were incubated inside a custom-made flow-through aquarium through which lake water from 42 m water depth in Lake Lucerne (~9 °C) was constantly pumped at 80 l/h (Figure 1 and Figure S1). The inorganic tracer bromide ($Br^-$) was injected to the inflow at a constant rate using a peristaltic pump (final concentration: 500 μM) and used to track ventilation. Before starting the experiments, luminophore beads were added to the sediment surface to trace sediment reworking [38].

There were six replicate cores for each experimental unit. Hypoxic controls consisted of sediment cores with core caps. For all oxic treatments caps were removed to allow for exchange with $O_2$-rich lake water. Chironomid larvae treatments (L) treatments were inoculated with 30 chironomid larvae from Lake Lucerne (~1700 individuals per $m^2$), whereas worm (W) treatments were inoculated with 150 oligochaetes from Lake Baldegg (~8500 individuals per $m^2$). These macrofaunal abundances corresponded to ~150% of the natural abundances at the sampled stations [37]. Sampling was performed after 6 (T1), 14 (T2), 27 (T3), 43 (T4), 63 (T5), and 82 d (T6). At each timepoint one core per treatment was destructively sampled. To minimize spatial biases, cores were arranged randomly inside the aquarium.

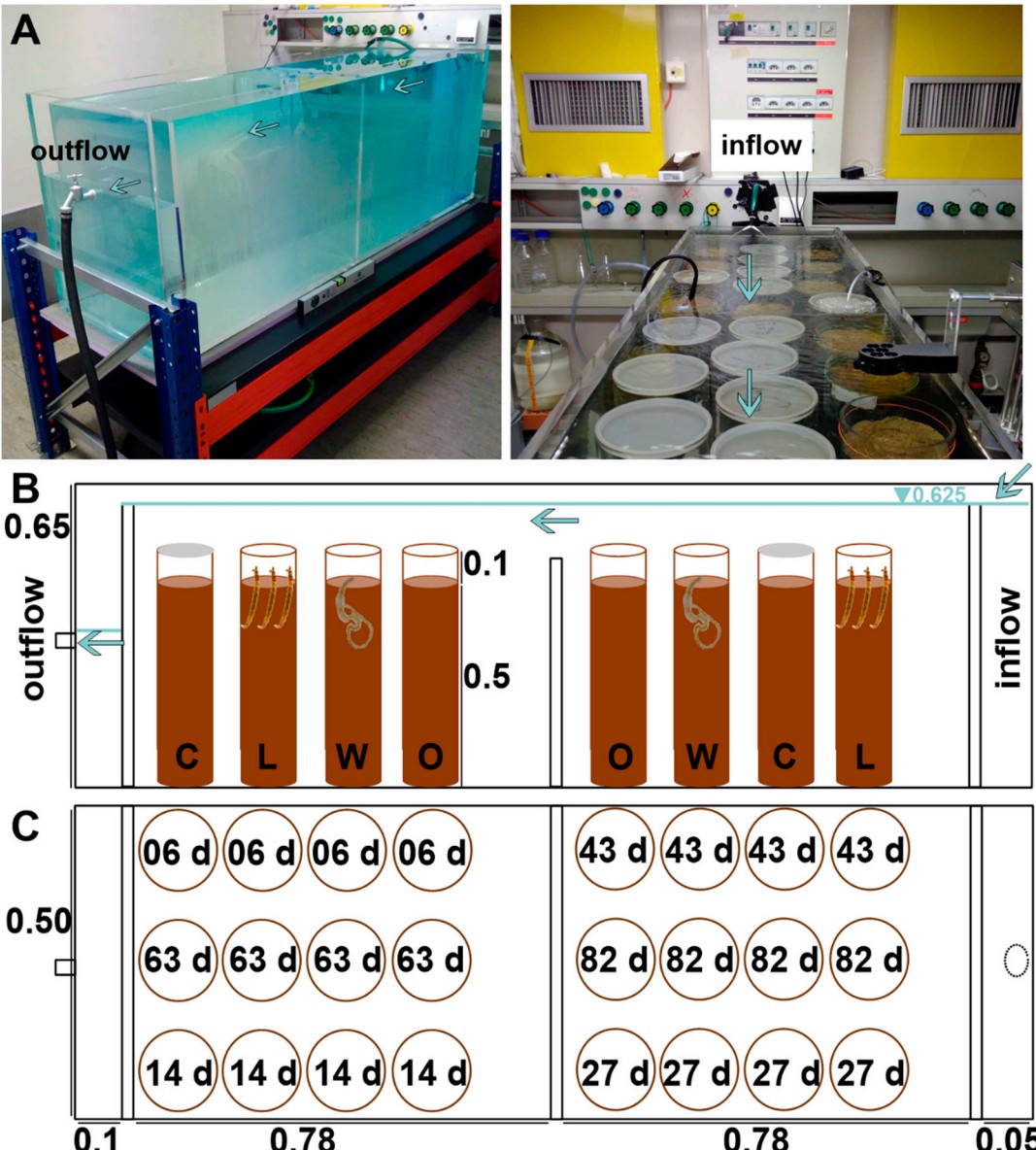

**Figure 1.** Scheme of the experimental setup. (**A**) Pictures of the mesocosm, flow direction is indicated by the arrows; (**B**) scheme of applied treatments, hypoxic control (C), chironomid larvae (L), oligochaetes (W) and oxic treatment (O). (Note: cores for each sampling time point are randomly placed and arranged in a row, in total three rows were placed in the tank; the blue line indicates the water surface); (**C**) arrangement of time points (time of sampling indicated by days (d)). In panels (**B**,**C**) total height (0.65 m), water level (0.625 m), height of core liners (0.6 m), height of cores (~0.5 m), overlying water (0.1 m), total length (1.71 m) and width (0.5 m) of the mesocosm are indicated. Lake water enters through a narrow pre chamber (inflow) and leaves on the opposite site (outflow).

Due to decreases in chironomid larvae abundances at 14 and 27 d, where only 6–12 larvae were recovered in L treatments, we added additional larvae (~20) at day 31. In all subsequent timepoints we recovered >25 larvae per core during sampling. The initial declines might have been caused by nightly migrations into the water column [12] followed by washout, by mortality, including predation by *Procladius* sp. and *M. fehlmannii*, or by metamorphosis.

### 2.3. Monitoring and Sampling

The dynamics of pH, redox potential, and $O_2$ in the top 0–4 cm of sediment were monitored weekly and then bimonthly using microsensors. Additional measurements were performed 1–2 days prior to destructive sampling.

For pore water and sediment sampling, cores were taken out of the aquarium. Bottom water was sampled using a syringe, porewater from 0.25, 0.5, 0.75, 1, 1.5, 2 cm was sampled using microrhizons (0.2 µm pore size, Rhizosphere), and at 3, 4, 6, 8, 12, 16, 20 and 28 cm using normal rhizons (10 cm porous part, 0.2 µm pore size, Rhizosphere). All rhizons were inserted horizontally via predrilled holes (sealed by transparent tape) on the side of the core liner. To prevent air ($O_2$) contamination, microrhizons were flushed with nitrogen gas immediately before insertion, whereas for normal rhizons with 3-way stop cocks the initial 1 mL of pore water was discarded. Furthermore, starting at 43 d, modified normal rhizons, that had been sealed at the base with a 2-cm piece of shrink tubing, were used. These modified rhizons effectively prevented contamination of porewater samples with bottom water that in some cases had been drawn down between the liner and sediment core. All pore water samples were immediately fixed using 1 M HCl for cations ($Fe^{2+}$, $Mn^{2+}$, $NH_4^+$; final pH: 4–5) and 1 M NaOH for anions ($SO_4^{2-}$, $NO_3^-$, $PO_4^{3-}$, $Br^-$) and stored at $-20\,^\circ$C.

After pore water sampling, sediment 'subcores' were taken from the top 0–6 cm of sediment by vertically inserting 5 and 20 mL cut-off syringes. The 5 mL syringes were closed with parafilm and immediately frozen at $-80\,^\circ$C for later analyses of DNA/RNA, solid-phase Fe pools, and sediment solid phase redox state (i.e., electron donor capacity (EDC)/electron acceptor capacity (EAC)). The 20 mL syringes for analyses of luminophores and chlorophyll *a* (chl *a*) were frozen at $-20\,^\circ$C. Prior to laboratory analyses, frozen subcores were extruded and sectioned into their appropriate depth intervals. Deeper sediment layers (6, 8, 12, 16, 20, 28 cm) were stored in the same way, but were sampled by extruding sediment and horizontally inserting cut-off syringes at each target depth.

### 2.4. Analyses

#### 2.4.1. Microsensor Analyses

We determined the vertical distributions of $O_2$, redox potential, pH, and $H_2S$ (at 82 d only) with 100 µm tip size microelectrodes using a field multimeter system with a micromanipulator (Unisense, Aarhus, DK). We always measured two sets of cores (time points) at each monitoring or before each sampling. Per each core and time point, multiple (3–5) replicate profiles were measured to check for spatial heterogeneity. The pH ranged from 7.7 to 6.6 and had distinct, diagenetically controlled profiles with no local peaks indicative of active cable bacteria (profiles in Figure S2). $H_2S$ concentrations were below detection to very low (<3.5 µM) (detailed profiles in Figure S3).

#### 2.4.2. Pore Water Analyses

**[$Fe^{2+}$] and [$Mn^{2+}$]**. Pore water concentration profiles of $Fe^{2+}$ and $Mn^{2+}$ were measured to determine the distributions of microbial Fe and Mn cycling. In total, 2 mL of pore water, that had been fixed with 20 µL of 30% HCl (Suprapur, trace metal certified, Sigma-Aldrich, Darmstadt, Germany), were analyzed by Inductively Coupled Plasma-Optical Emission Spectroscopy (5100, Agilent Technologies, Santa Clara, USA) after dilution with Milli-Q water and compared to multi element standards (solution IV, MERCK, Certipur, Darmstadt, Germany).

**[$NH_4^+$]**. Concentrations of $NH_4^+$ were determined spectrophotometrically on 96 well plates according to Kempers and Kok [39]. The absorbance of blue indophenol complexes (modified Berthelot reaction) was measured in duplicates on a spectrophotometer (Synergy HT, BioTek photometer, Bad Friedrichshall, Germany) at 640 nm against a blank (MQ, pH 4). Standards and blanks were made using ammonium chloride (Sigma-Aldrich, analytical grade) dissolved in Milli-Q water.

**[$SO_4^{2-}$, $NO_3^-$, $NO_2^-$, $Br^-$].** Inorganic anion concentrations were measured on an ion chromatograph (IC, DIONEX DX-ICS-1000, Thermo Fisher Scientific, Waltham, USA) equipped with an AG14A guard column, an AS14A main column and a 4 mm suppressor (ADRS). The eluent was a solution of 8 mM $Na_2CO_3$ and 1 mM $NaHCO_3$ prepared on the day of the measurement. Standards were prepared for each ion separately by dissolving their sodium salts (all Sigma-Aldrich, analytical grade) in Milli-Q water. [$NO_2^-$] was negligible throughout the experiments and is not presented in the results.

2.4.3. Solid-Phase Analyses

**Bioavailable sedimentary iron.** For quantifications of biologically available particulate Fe(II) and Fe(III), frozen subcores were cut into 2 mm slices, immediately placed into 5 mL HCl (0.5 M), and weighed. After extraction (30 min, shaking) the extract was split into two subsamples. To determine Fe(II), 40 μL of extract were mixed with 2 mL of 0.02% ferrozine in 50 mM HEPES at pH 7 [40] and then quantified photometrically on a Plate Reader (Synergy HT, BioTek). For total Fe, 1 mL of extract was mixed with 0.2 mL hydroxylamine (1.5 M) to reduce Fe(III) to Fe(II) and then measured as above. Fe(III) concentrations were the difference of total Fe and Fe(II). Standards consisted of dilution series of 100 mM $FeSO_4$ in 0.5 M HCl. To get insights into more crystalline Fe phases, sediment was sequentially also extracted using 3M HCl for 24 h at 60 °C for cores sampled after 27, 63, and 82 d. Afterwards, extracts were pH corrected using NaOH and analyzed as above, except that the reducing agent was 6.25 M hydroxylamine.

To determine changes in the redox state of the sediment with depth, we quantified the electron donating capacity (EDC) and electron accepting capacity (EAC) using mediated electrochemical analysis [41,42]. Further information about the experimental setup and analysis are given in Text S1.

Please see Text S1 for further information.

**Luminophore beads.** To quantify macrofaunal sediment reworking, luminophore beads consisting of fluorescent paint-covered silt particles with 30–40 μm diameter (D50: 30–40 μM, Environmental Tracing, Helensburgh, UK) were added at the start of the experiment. Depth distributions of beads (0.25, 0.5, 0.75, 1, 1.5, 2, 2.5, 3, 3.5, 4, 4.5, 5, 5.5, and 6 cm) were analyzed after 6, 43, 63 and 82 d. Frozen sediments were extruded from 20 mL cut-off syringes and sliced into 2 mm depth intervals. Samples were weighed, dried at 50 °C, and homogenized. Afterward each sample was placed in a plastic petri dish and distributed evenly by gently shaking until a monolayer of luminophore beads was observed. Photographs of the whole petri dish were taken using a Sony 6000a camera under UV light and all luminophores on the pictures were counted by eye in Image J. For each petri dish three pictures were taken, with petri dishes being shaken between pictures.

**Chlorophyll a.** Sediment chl a was quantified after 82 d as a proxy for the content of fresh algal organic carbon (OC). Chl a was extracted from ~0.3 g of wet sediment in the dark for 20 h with 90% acetone (2.5 mL) at −20 °C [43], and quantified by spectrophotometry (UV-VIS, VARIAN, Cary, 50 Scan, Agilent Technologies, Santa Clara, SC, USA) before and after acidification according to Lorenzen [44] to quantify contents of chl a and pheopigments. Ratios of chl a to pheopigments were used as a proxy for OM freshness.

**Nucleic acid extraction from sediment.** DNA and RNA were extracted simultaneously following lysis protocol I in Lever, et al. [45] using 100 μL of 10 mM sodium hexametaphosphate to prevent adsorptive losses. The detailed extraction protocol can be found in the, Texts S2 and S3.

**DNA extraction from macrofauna.** Prior to extraction, macrofaunal samples (whole specimens of larvae and worms) were cleaned with molecular grade water and cut into pieces using an ethanol wiped and flame sterilized scalpel. From a subset of specimens the gut was separated from the rest of the body using hypodermic sterile needles and extracted separately (see also [37]). Macrofaunal DNA was extracted following lysis protocol I in Lever, et al. [45]. For further details see Text S2.

**Quantitative polymerase chain reaction (qPCR).** For methodological information on the reverse transcription and quantification of rRNA gene copy numbers, we refer the reader to section Text S4.

**Next Generation Sequencing (NGS).** NGS was performed on a MiSeq (Illumina Inc., USA) using 10% PhiX. For 16S, the universal prokaryotic 16S rRNA gene primer pair Univ519F (CAG CMG CCG CGG TAA) and Univ802R (TAC NVG GGT ATC TAA TCC) [46,47] was used. For details of library preparation please see Text S5.

### 2.4.4. Bioinformatic Analyses

A first quality check with FastQC was performed followed by trimming read ends and merging the pairs into amplicons using the software seqtk and flash (N:103, usearch v11.0.667_i86linux64, Trim R1: 15, Trim R2: 20, FLASH v1.2.11, Min Overlap: 15, Max Overlap: 300, Max Mismatch Density: 0.25), data were dereplicated to obtain unique amplicons using usearch:fastx_uniques. Primer sites were trimmed using usearch (usearch v11.0.667_i86linux64, Amplicon range: 100–600, Number of mismatches: 1, Coverage: full-length, Wildcards enabled: IUPAC codes). Afterwards a quality filtering step using prinseq (PRINSEQ-lite 0.20.4, Size Range: 100–600, GC Range: 30–70, Min Q Mean: 20, Number of Ns: 0, Low Complexity: dust/30) was performed. Then ZOTUs were generated using usearch (UNOISE3), 97% clustering was performed (UPARSE), and a denoising (error correction) was performed using usearch:unoise3. Amplicons were mapped to ZOTUs to generate count tables using usearch:otutab. Bacterial phylogenetic assignments were done with SINTAX using the SILVA_128 database (v11.0.667_i86linux64). Archaeal 16S rRNA gene assignments were done in ARB (www.arb-home.de) by neighbor-joining phylogenetic trees. The ARB database was based on a SILVA 16S database with manually optimized alignments that had been expanded with 16S gene sequences from whole-genome studies and updated to state-of-the-art phylogenetic nomenclature. All further analyses (NMDS, CAP) were performed using RStudio under R version 3.5.2 and packages phyloseq, vegan, and ggplot2 [48].

### 2.4.5. Statistical Analyses

Pairwise comparisons between porewater-dissolved and solid-phase chemicals were performed based on Wilcoxon signed-rank tests to assess statistical differences between treatments.

## 3. Results

### 3.1. Visual Observations

The sediment initially retrieved from Lake Zurich was dark brown and clearly laminated in the upper 10 cm. With time, the surface turned to a lighter brownish color except in hypoxic control (C) treatments. This color change reached deeper layers and was more blurry in W ($\geq$2 cm) compared to O or L treatments (0.75–1 cm) (Figure 2d,e). In W treatments a dense network of tunnels established to 20 cm within <7 days after the start of the experiment. Worms were often observed at the sediment surface, with their heads buried and tail ends actively undulating above the sediment water interface (Figure 2c). Fine, filamentous worm feces accumulated on the sediment surface over time (Figure 2a). After addition to cores, larvae quickly buried themselves into the sediment and subsequently remained hidden. Diverse burrow and tube types were formed, confirming the presence of diverse chironomid communities. Some burrows were only visible as holes from the sediment surface. Others had tubes (chimneys) that extended 1–2 cm above the sediment surface (Figure 2h). Ventilated, U-shaped burrows with a lighter color indicating sediment oxidation were also observed (Figure 2f). Larval feces consisted of roundish pellets that were deposited on the sediment surface (Figure 2b).

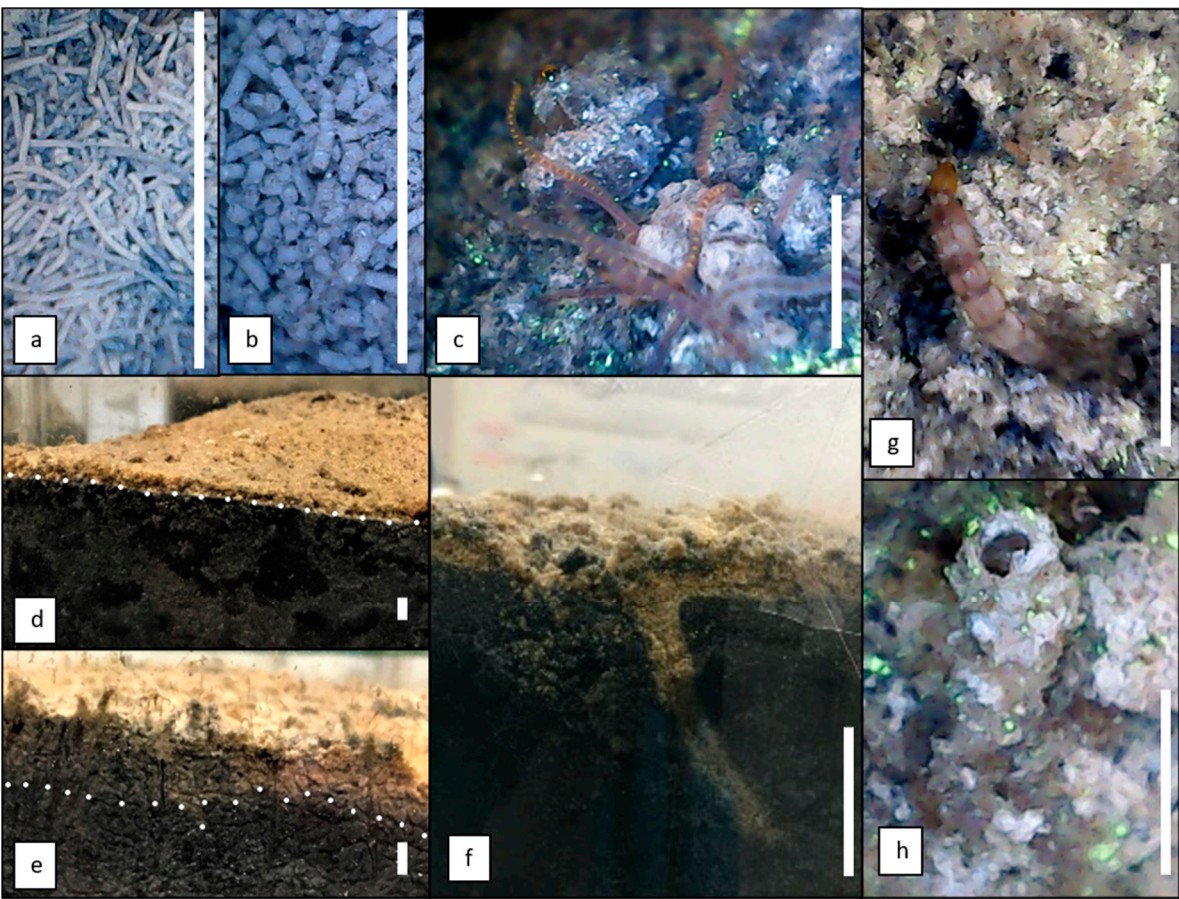

**Figure 2.** Images of fecal pellets of oligochaete worms (**a**) and chironomid larvae (**b**) on the sediment surface. Oligochaete worms at the surface, actively moving (**c**). Interface of oxidized to reduced sediment in an oxic treatment (**d**) and W treatment (**e**) (interface indicated by white dotted line). Ventilated U-shaped larval burrow (light color indicates oxidized sediment) (**f**) and tube entrance at the sediment surface (**h**). Larvae on the sediment surface immediately after addition to the cores (**g**). The white bars serve as reference scale (length = 1 cm).

### 3.2. Microsensor Measurements of $O_2$ Concentrations and Redox Potential

Average $O_2$ microprofiles for 27, 63, and 82 d are shown in Figure 3 (for individual profiles and all time points, see Figure S4). $O_2$ microprofiles confirm the hypoxic, perhaps even anoxic, conditions in hypoxic controls. The $O_2$ at low concentrations in bottom water was likely introduced during the opening of cores for measurements. Across all oxic treatments, $O_2$ concentrations in the bottom water are between 240 and 260 $\mu$M. The $O_2$ penetration depth increases over time from ~3.4 to ~8.2 mm and was similar across oxic treatments, except that $O_2$ penetration depth was significantly ($p < 0.05$) lower in W compared to O and L treatments after 82 d. $O_2$ concentrations above the sediment surface, moreover, vary more between replicates in W compared to O and L treatments (Figure S5).

Redox potential ($E_h$) values ranged from +600 mV at the surface to −200 mV at 4 cm in all oxic treatments (Figure 4). $E_h$ values in the C treatment showed reducing conditions with all values <40 mV. $E_h$ values in O, L and W treatments ranged from +400 to +600 mV at the sediment surface. In O treatments, $E_h$ decreased with depth to ~0 mV. The $E_h$ in L and W treatments also decreased with depth, but more gradually. After 82 d, W treatments showed significantly ($p < 0.05$) higher $E_h$ values below 2 cm compared to L and O treatments.

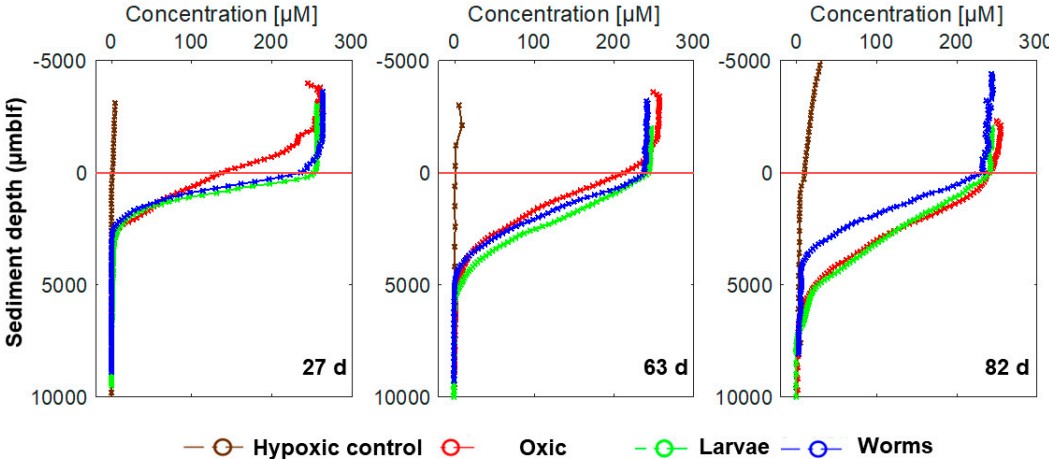

**Figure 3.** $O_2$ microsensor profiles vs. depth in micrometer below lake floor (µmblf). The depth of the sediment–water interface was set to 0 (horizontal red line). Profiles of Oxic, Larvae, and Worm treatments after 63 and 82 d. The profiles are averages of five replicate profiles, whereas those after 27 d are averages of three replicate profiles. Only one profile was measured in hypoxic controls. Individual profiles and additional dates can be found in Figure S4.

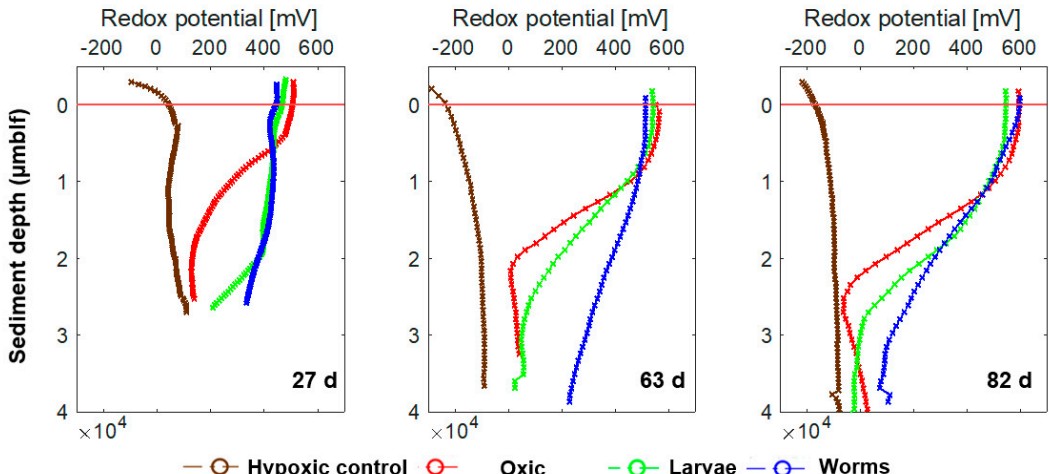

**Figure 4.** Redox microsensor profiles obtained during the experiment. The horizontal red line indicates the sediment–water interface (SWI). Profiles from 63 and 82 d (O, L, W) are averaged over three replicate profiles; those from 27 d and all controls are based on a single profile. Complete datasets of redox potential ($E_h$) with individual profiles can be found in Figure S6.

### 3.3. Pore Water Concentrations of Anions and Cations

Despite showing considerable fluctuations with depth, which could in part result from heterogeneous distributions of macrofauna and their activities, porewater geochemical profiles are influenced in consistent ways by the presence of $O_2$ and different macrofaunal groups.

### 3.3.1. Bromide ($Br^-$)

$Br^-$ was absent from controls as these treatments were kept closed throughout the experiment. O, L, and W treatments have similar bottom water $Br^-$ concentrations (~400 µM) throughout the experiment, an indication that [$Br^-$] was constant in tank water (Figure 5). Over time, $Br^-$ penetration depths in O, L and W treatments increased from 8 (6 d) to >20 cm (82 d). Matching known bioturbation activities, $Br^-$ penetration into deeper layers is fastest in L and slowest in O treatments, as evidenced by significantly ($p < 0.05$) higher $Br^-$ concentrations in L compared to O treatments.

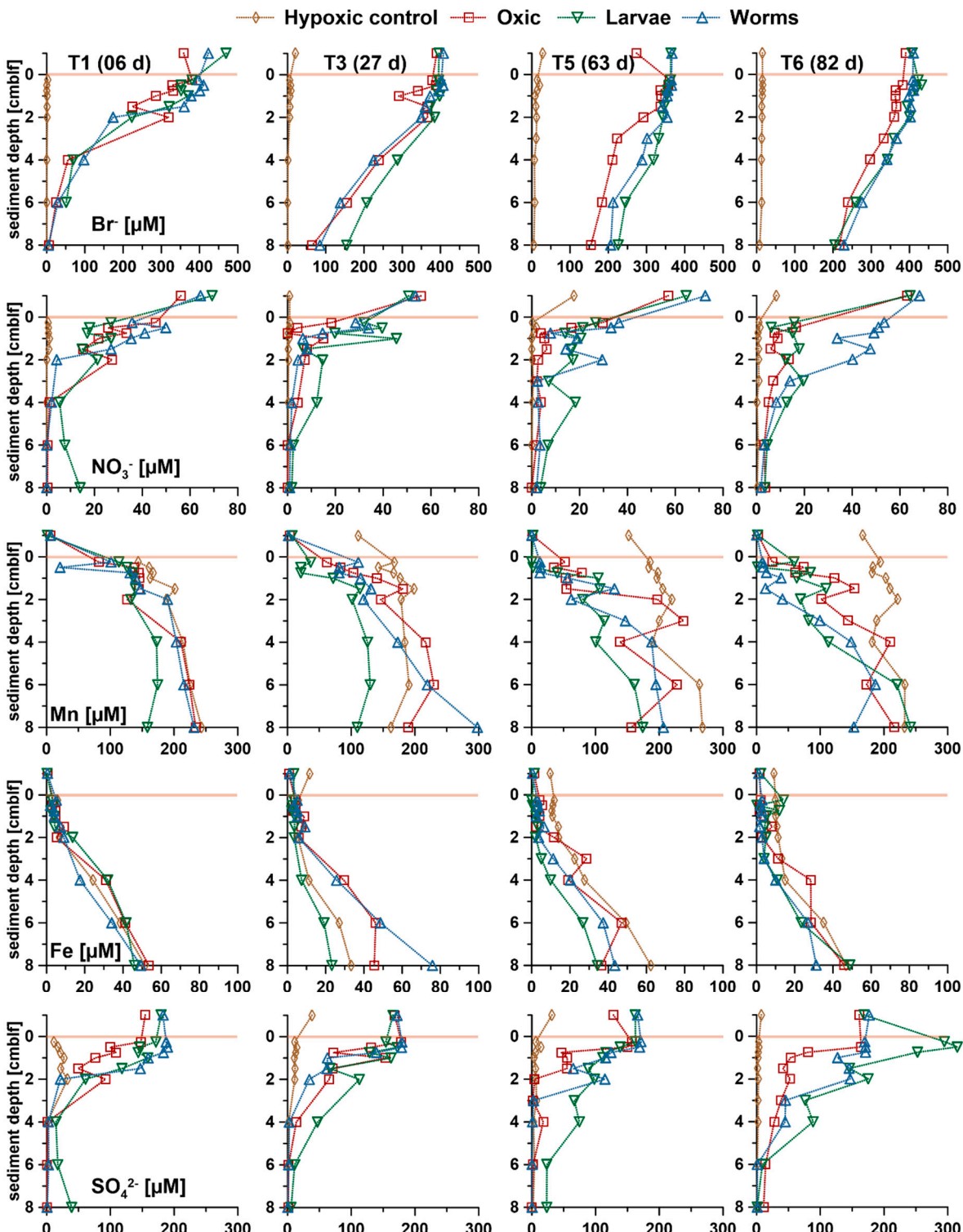

**Figure 5.** Concentration profiles of the measured pore water species in the first 8 cm of sediment (note that the concentration ranges (x-axis) change with analytes, cmblf (centimeter below lake floor). The sediment–water interface is indicated with a horizontal line. A plot with all measured timepoints can be found in Figure S7.

### 3.3.2. $NO_3^-$ and $SO_4^{2-}$

$NO_3^-$ was nearly absent from hypoxic controls, whereas all oxic treatments had significant $NO_3^-$ concentrations in bottom water (49–73 µM). Compared to O treatments, L treatments had significantly higher $NO_3^-$ concentrations between 2 and 8 cm. W also

had elevated $NO_3^-$ concentrations after 82 d, but only in the top 2 cm. In hypoxic controls, bottom water $[SO_4^{2-}]$ decreased from 97 µM at the beginning to 7 µM after 3 months, whereas $[SO_4^{2-}]$ remained at ~160 µM in all oxic treatments. Similar to $[NO_3^-]$ treatments, $[SO_4^{2-}]$ in larval treatments was significantly ($p < 0.05$) elevated between 2 and 8 cm compared to W and especially O treatments (Figure 5).

### 3.3.3. $Mn^{2+}$ and $Fe^{2+}$

Dissolved $Mn^{2+}$ and $Fe^{2+}$ were present in bottom water of controls, but close to 0 µM in bottom water of all oxic treatments. In all treatments, $[Mn^{2+}]$ increased to ~15 cm (~250 µM) and stayed constant or decreased slightly below, whereas $[Fe^{2+}]$ increased consistently below 2 cm (Figure 5). Compared to O treatments, $Mn^{2+}$ and $Fe^{2+}$ concentrations were significantly ($p < 0.05$) lower from 2–8 cm in L treatments. This difference was most pronounced during the first two months.

### 3.3.4. $NH_4^+$

$NH_4^+$ was present in bottom water of controls but not oxic treatments, which also had lower concentrations in the top 2 cm (Figure S8). While $[NH_4^+]$ below 2 cm overlapped and fluctuated significantly with depth in all treatments, L treatments had significantly ($p < 0.05$) lower $[NH_4^+]$ throughout the experiments than all other treatments. W treatments only showed slightly (not significant, $p = 0.22$) lower $[NH_4^+]$ compared to O at later timepoints (>63 d).

### *3.4. Solid Phase Analyses*

### 3.4.1. Iron and EAC/EDC

Depth profiles of the % contribution of Fe(III) to total bioavailable iron Fe(II), all extracted with 0.5 M HCl, are shown in Figure 6 (for contents of bioavailable Fe(II) and Fe(III) for all time points see Figure S9). Fe(II) dominated everywhere except in surface sediments of oxic treatments at later time points. Despite the higher redox potential of W treatments there were no clear differences in Fe(III) trends between O, L, and W treatments. Additionally, EDC analyses confirm the increasingly reducing conditions with sediment depth, and the lower ratio of EDC:EAC confirms the more oxidizing conditions in surface sediment—again with no differences between O, L, and W treatments (Figure S10).

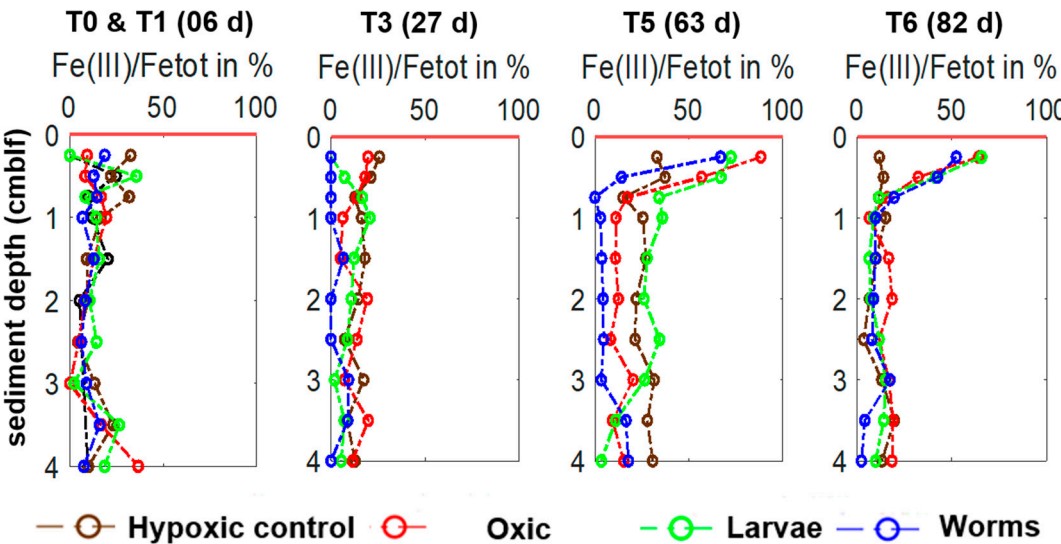

**Figure 6.** Bioavailable Fe (III) of total Fe (extracted with 0.5 M HCl). For T0 and T1 (6 d) only one profile was measured, for T3 (27 d), T5 (63 d), and T6 (82 d) profiles are averages of two profiles per core.

### 3.4.2. Luminophores and Chlorophyll a

Luminophore counts and Chl *a* data suggest minimal sediment reworking by fauna and minimal differences in degradation rates of labile organic matter between treatments (Figures S11 and S12).

### 3.5. Microbial Community Composition

### 3.5.1. qPCR Values of Bacterial and Archaeal 16S rRNA Genes

Bacterial and archaeal gene copies showed little systematic variation with depth and over time (Figure S13). Bacterial copy numbers varied around $2 \times 10^9$ copies per $g^{-1}$ wet sediment. The only noteworthy change was a decrease in gene copies in C and W treatments after 82 d. Archaeal gene copies showed the same pattern but had a lower average abundance (~$1 \times 10^8$ copies per $g^{-1}$ sediment). In addition, Bacteria-to-Archaea Ratios (BARs) increased over time in surface sediments of oxic treatments. This trend was also present in BARs that were calculated based on reverse-transcribed RNA copies (Figure S13).

In addition to these treatment-related sedimentary trends, we examined BARs in whole specimens, fecal pellets, and tubes/burrow walls of larvae and worms (Figure 7). Sediment BARs showed the lowest ratios (24.2 ± 24.3, n = 108). Worm specimens (185.3 ± 93.2, n = 10), tubes (187.0 ± 132, n = 13), and feces (176.8 ± 121.6, n = 9) had highly similar BARs that were on average 8–10 fold higher than bulk sedimentary BARs. Larval specimens (238.0 ± 161.3, n = 13) and feces (211.6; 256.5, n = 2) had similar BAR values to worm specimens and feces, however, larval tubes (53.2 ± 28.5, n = 6) had significantly lower BAR values that were intermediate between those of sediments and larval specimens and feces.

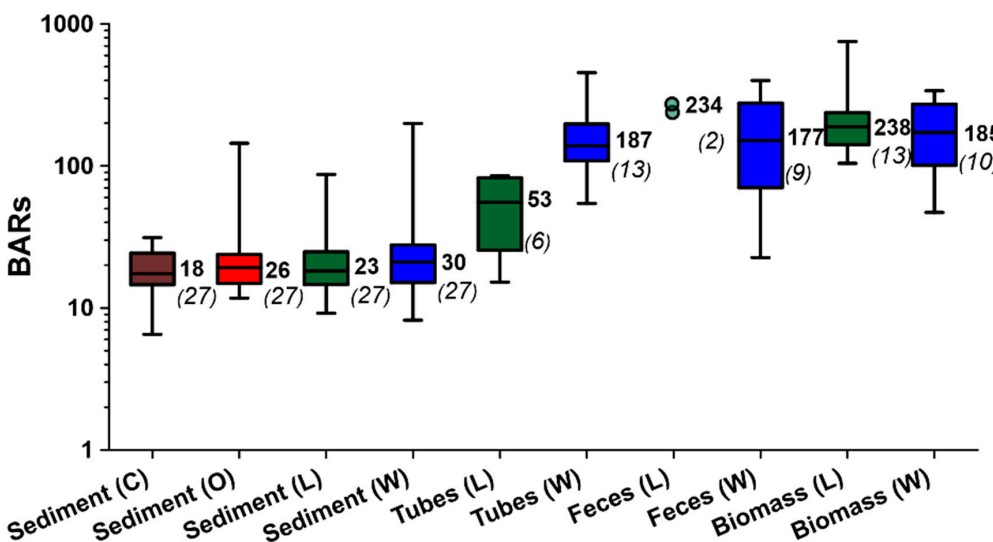

**Figure 7.** Boxplots of Bacteria to Archaea Ratios (BARs) in worms, larvae, worm and larval tubes and fecal pellets, and bulk sediments from 0 to 4 cm (T3 (27 d), T5 (63 d), T6 (82 d)) for hypoxic control (C), oxic (O), worm (W) and larvae (L) treatments. Numbers in bold indicate mean values, numbers in parentheses indicate sample sizes. Please note for larvae fecal pellets only two samples were found, therefore they are displayed as two individual circles.

### 3.5.2. Amplicon Sequencing of 16S rRNA

An overview of the bacterial and archaeal read numbers (NGS) and zOTU numbers can be found in Table S1.

### 3.5.3. Sediment Microbial Community Zonation Patterns across Experimental Treatments

Bacterial and archaeal communities in hypoxic controls were highly similar to those recently reported for the same location [49] throughout the experiment but changed significantly in surface sediments of experimental treatments. Since bacterial and archaeal RNA

sequence compositions were highly similar to archaeal 16S rRNA gene sequence compositions (Figures S14 and S15), we only discuss 16S rRNA gene sequence data in the following text.

### 3.5.4. Bacteria

NMDS analyses showed that at the end of the experiment bacterial communities in surface sediment (upper 1.5 cm) of all oxic treatments cluster away from the hypoxic control (Figure 8A). This difference was absent from deeper samples, where communities in all four treatments follow the same depth-related trend, indicating no impact on $O_2$ or bioturbation. Surface sediments of oxic treatments mostly aligned with concentrations of $O_2$ and other electron acceptors ($SO_4^{2-}$, $NO_3^-$), as well as redox potential (Figure S16). This suggests that changes in O, L, and W treatments are mainly driven by oxygenation and/or oxidation of surface sediment. In deeper sediment layers, bacterial communities correlated mostly with $Fe^{2+}$ and other reduced compounds ($NH_4^+$ and $Mn^{2+}$). Phylogenetic comparisons of bacterial communities across the four treatments (Figure 9; for details see Text S7) suggest that among the dominant phyla (*Proteobacteria*, *Bacteroidetes*, *Actinobacteria*, *Planctomycetes*, *Verrucomicrobia*, *Chloroflexi*, *Omnitrophica*, *Cyanobacteria*, *Latescibacteria*) there was a gradual increase in $\alpha$-(mostly *Caulobacterales*), $\beta$-(*Nitrosomonadales*, *Burkholderiales* and *Rhodocyclales*) and $\gamma$-(*Methylococcales, Cellvibrionales*) *Proteobacteria* in O, L, and W treatments through time. These increases were accompanied by decreases in $\delta$-*Proteobacteria* (mostly *Synthrophobacterales*), *Actinobacteria* (*Actinobacteria, Acidimicrobiia, Thermoleophilia*), *Planctomycetes* (*Planctomycetacia*), *Chloroflexi* (*Dehalococcoidi, Anaerolineae*), *Omnitrophica*, *Latescibacteria*, and *Cyanobacteria* (*Cyanobacteria*, chloroplasts). By comparison, bacterial communities in hypoxic controls changed little with depth or over time.

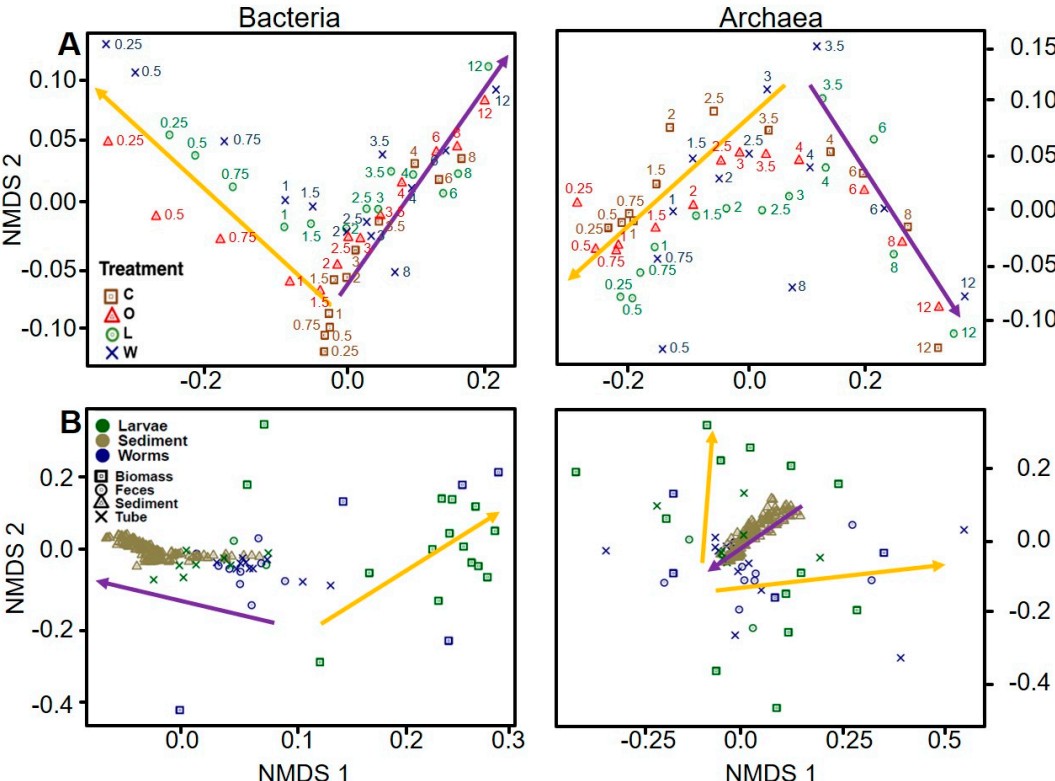

**Figure 8.** NMDS ordination analysis for bacterial (**left**) and archaeal (**right**) ZOTUs based on Bray–Curtis method. (**A**): Shown are T6 (82 d) sediment samples from all layers and treatments, with two manually placed arrows pointing out general trends according to sediment depth. Sample depths in cm are shown next to each data point. (**B**): Includes macrofaunal features (feces, biomass, and tube) with bulk sediment samples. The arrows where placed manually to highlight the clear separation of all macrofaunal features from sediment samples, especially those of larvae and worm biomass.

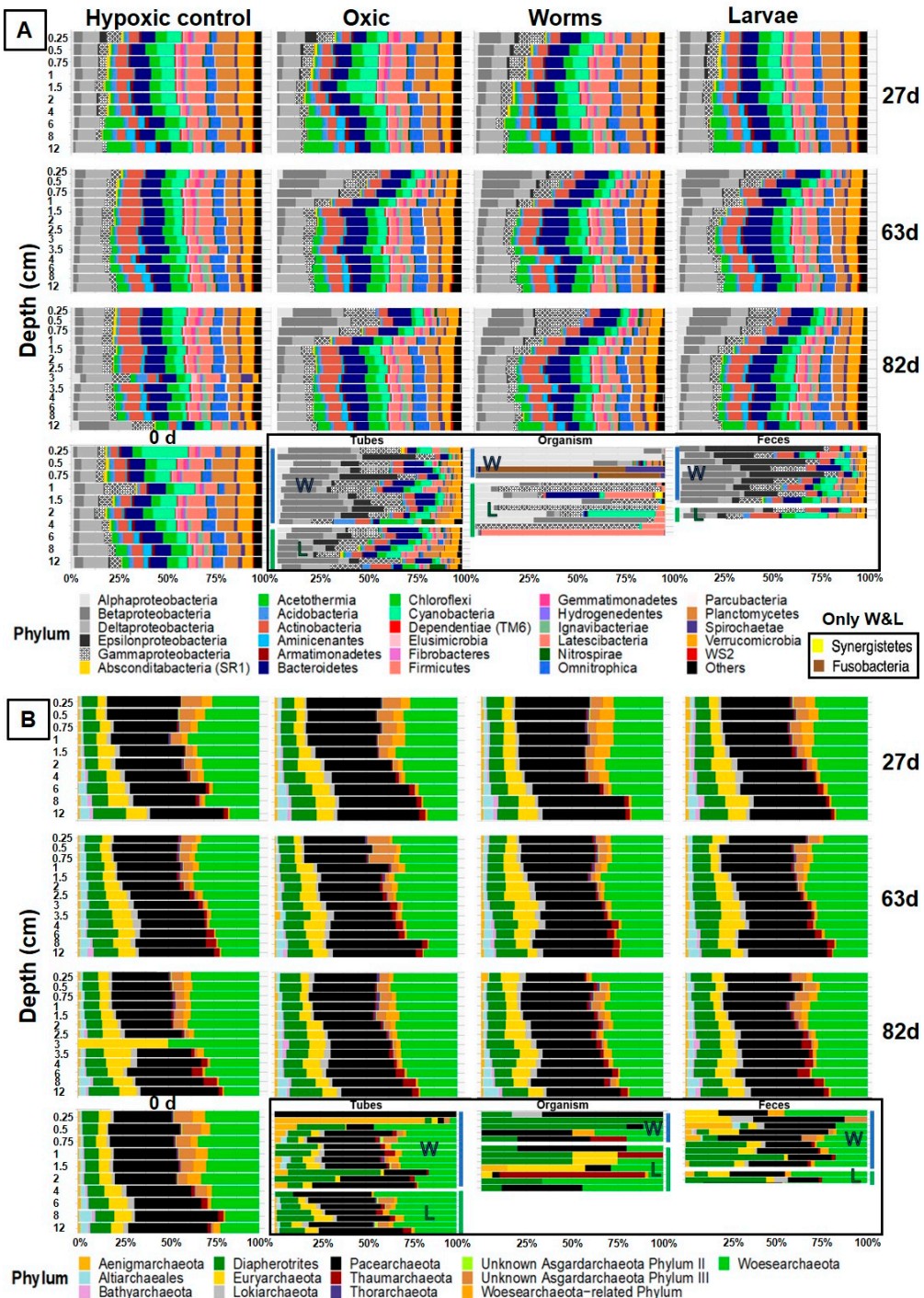

**Figure 9.** Relative abundances (in %) of Bacteria (**A**) and Archaea (**B**) on the phylum level vs. sediment depth (cm) (note: Proteobacteria shown at class-level). Columns show different treatments (hypoxic control, oxic, worm and larvae treatments), rows show timepoints: T3 (27 d), T5 (63 d) and T6 (82 d). Note: last row shows T0 (0 d) and different macrofaunal features (tubes of worms and of larvae, whole organisms, fecal pellets of worms and larvae, all timepoints). All samples belonging to worms are marked with "W" and a blue line and all samples belonging to larvae are marked with "L" and a green line. Low abundant phyla (total relative abundance < 0.25% or maximum relative abundance < 1%) were merged together into "Others".

### 3.5.5. Archaea

While archaeal communities also clustered according to sediment depth, no clear differences between the four treatments can be observed in NDMS plots at the end of experiments

(Figure 8A). This is in line with CAP analyses, which showed that archaeal community composition does not strongly correlate with measured chemical data (Figure S16). Archaeal community depth profiles were stable through time (Figure 9; for details see Text S7) and indicate dominance of *Pacearchaeota*, *Woesearchaeota*, *Diapherotrites* and *Euryarchaeota*. While *Pacearchaeota* showed almost no depth-related changes, relative abundances of *Woesearchaeota* decreased slightly below 4 cm, whereas those of *Altiarchaeales* and *Euryarchaeota* increased with depth. Two uncharacterized phylum-level groups, which we call Unknown Asgardarchaeota Phylum III and *Woesearchaeota*-related phylum, had elevated abundances in surface sediments.

3.5.6. Comparison of Microbial Communities in Sediment, Faunal Specimens, Burrow Structures, and Feces

NMDS analyses indicate strong community differences between sediments and macrofaunal biomass, feces, and burrow structures (Figure 8B), with macrofaunal biomass representing a variable but distinct end member for larvae and worms in Bacteria and for larvae only in Archaea. Bacterial communities in burrow structures and feces clustered with or between macrofaunal biomass and oxic surface sediments in Bacteria. This pattern was also evident in archaeal communities from larval burrows and feces. By contrast, archaeal communities from worm biomass, burrows, and feces were strongly scattered and showed no clear clustering patterns.

Bacterial communities in tubes, feces, and faunal specimens at higher phylogenetic levels were also different from those in sediment and showed general trends that were similar to those in NMDS plots (Figure 9). Relative abundances of $\beta$-, $\varepsilon$- and $\gamma$-*Proteobacteria* were clearly higher, whereas contributions of *Omnitrophica* and *Spirochaetes* were lower in tubes of larvae and worms compared to surface sediments. Worm and larval feces were similar to worm and larval tubes, respectively, but had higher fractions of $\varepsilon$-*Proteobacteria*. Whole worms and larval samples had very different, less diverse and more variable bacterial DNA sequences than the other sample categories. Whole worms contained mainly $\alpha$-, $\beta$-*Proteobacteria* and/or *Fusobacteria*, while whole larvae were dominated by $\alpha$-, $\beta$- and/or $\gamma$-*Proteobacteria* or *Firmicutes*.

Archaeal communities of fauna, burrow structures, and feces were dominated by few archaeal taxa (Figure 9). Larval and worm tubes and feces were, similar to sediments, dominated by *Diapherotrites*, *Pace-* and *Woesearchaeota*. Archaeal communities of whole organisms were highly variable, with dominant members including, *Pace-* and/or *Woesearchaeota* in worms, and *Diapherotrites*, *Eury-*, *Pace-*, *Thaum-*, or *Woesearchaeota* in larvae.

## 4. Discussion

We investigated the effect of bottom water oxygenation and chironomid larval and oligochaete worm colonization in previously macrofauna-free, hypoxic sediment to gain a better understanding of how lake oligotrophication and bottom water reoxygenation affect biogeochemical processes and microbial communities in lacustrine surface sediment. We found that the presence of $O_2$ alters pore water and solid phase geochemical conditions in surface sediments and leads to a gradual but clear change in microbial community composition over time. We, moreover, showed that the presence of macrofauna adds to the effect of $O_2$ conditions, but was generally weak and more strongly influences biogeochemical parameters than microbial community composition. Larvae and worms had distinct effects on sediment biogeochemistry with larvae mainly altering pore water geochemical gradients, whereas worms raised the redox potential of the sediment. While differences in the microbial community composition were very small between O, L and W treatments, communities of microorganisms were highly distinct from sediments in macrofaunal feces, tubes, and biomass. In the following, we discuss the influences of $O_2$ and the added effects due to ventilation of larvae and reworking by worms on sediment biogeochemistry and the microbial community structure, as well as possible reasons for the observed weak treatment effects of macrofauna addition.

### 4.1. The Effect of Bottom Water $O_2$ and Macrofaunal Activity on Sediment Biogeochemistry

The presence of $O_2$ in sediment-overlying water was the main driver of changes in pore water and solid-phase biogeochemistry in our experiment. In hypoxic controls, the oxic lake water-derived electron acceptors $NO_3^{2-}$ and $SO_4^{2-}$ were strongly depleted relative to oxic treatments. Similarly, concentrations of redox-sensitive reduced solutes, $Fe^{2+}$ and $Mn^{2+}$ (Figures 3 and 5), which were produced in the sediment cores through microbial Fe(III) and Mn(IV) reduction and were nearly absent in bottom water of oxic treatments, accumulated in water overlying hypoxic controls. The strong impact of $O_2$ was also evident from lower redox potentials (Figure 4), Fe(III) contents (Figure 6), and EACs (Figure S10) in surface sediments of controls compared to oxic treatments.

By comparison, the biogeochemical impact of macrofauna was minor. In contrast to other studies we observed no decreased $O_2$ penetration (Figure 3), which previous studies on chironomid larvae have interpreted to indicate increased aerobic microbial activity [23,50,51]. Presence of larvae elevated dissolved electron acceptor concentrations ($NO_3^-$ and $SO_4^{2-}$), especially from 2 to 6 cm. These increases were likely due to ventilation activities which led to bioirrigation and increased nitrification and S oxidation through the input of $O_2$ (Figure 5). Bioirrigation was likely also responsible for the decreased concentrations of reduced compounds ($Fe^{2+}$, $Mn^{2+}$, $NH_4^+$) in the same layers (Figure 5). Bioirrigation likely diluted these compounds with compound-depleted bottom water and stimulated in situ chemical and biological oxidation with $O_2$ and possibly $NO_3^-$.

The impact of chironomid larvae on porewater chemistry in the bioirrigated layer is supported by ratios of pore water $[NO_3^-]$, $[SO_4^{2-}]$, $[Mn^{2+}]$, $[Fe^{2+}]$, $[PO_4^{3-}]$ and $[NH_4^+]$ to the conservative tracer $[Br^-]$. These indicate increased ratios of $[NO_3^-]$ and $[SO_4^{2-}]$ to $[Br^-]$, and decreased ratios of $[Mn^{2+}]$ and $[Fe^{2+}]$ to $[Br^-]$ in L treatments compared to O and W treatments from 2 to 8 cm (Appendix A, Figure A1). The absence of a clear larval impact in the top 2 cm is likely related to dominance of diffusive mixing and advection driven porewater exchanges with bottom water in this surface layer.

Compared to larvae, worms had an overall lower biogeochemical impact on surface sediments that mainly manifested itself in an increase in redox potential. This increase in redox potential was likely related to reworking through the observed head-down, bottom-up conveyor feeding, which visibly oxidized the top $\geq 2$ cm of sediment. Similar increases redox potential have been previously noted for bioturbated treatments with the tubificids *Tubifex spp.* and *C. riparius* [35]. Yet, it remains unclear which chemical transformations are responsible for these increases in redox potential in W treatments. Clear shifts in measured iron pools from Fe(II) toward Fe(III) were absent (Figure 6), Figure S9), suggesting that—if significant oxidation of solid-phase Fe(II) occurred—it took place in (labile) iron pools that made up minor portions of bulk iron extracts. Even though it is known that sedimentary OM can also act as a redox-active phase [52,53], changes in EAC and EDC measurements do not indicate a significant impact of worms on the redox potential of organic compounds (Figure S10).

Notably, despite the increased redox potential and visible oxidation of the top $\geq 2$ cm of sediment, luminophore counts suggest no significant reworking in W treatments. It is possible that reworking was mainly confined to the surface centimeter, though this does not explain the observed redox potential and color changes. Alternatively, the constantly observed, undulating movements of worm tails above the sediment surface may have significantly increased input of oxic overlying water into the top 1–2 cm of sediment and thus driven the observed changes in redox potential. Such locally enhanced exchanges between surface sediments and overlying water were, however, not visible in porewater concentrations of electron acceptors or reduced species. Thus, as for larvae, clear impacts of worm bioirrigation on porewater geochemistry in the top layer of sediment may have been masked by significant diffusive and advection-driven mixing with bottom water.

### 4.2. Effect of Bottom Water $O_2$ on Sediment Microbial Communities

Reflecting the fact that the biogeochemical impact of $O_2$ was strongest in surface sediments, microbial community changes in response to $O_2$ mainly occurred within the top

2 cm of sediment. Hereby statistical tests suggest changes in $O_2$, $NO_3^-$, $SO_4^{2-}$, and $Fe^{2+}$ concentrations and redox potential as likely drivers of microbial community changes (Figure S16). $O_2$ presence can explain the increased relative abundances of chemoorganotrophic aerobes, such as *Caulobacterales* (*α-Proteobacteria*), *Cellvibrionales* (*γ-Proteobacteria*) or *Sphingobacteriia* (*Bacteroidetes*) [54–57] and aerobic methanotrophs (*Methylococcales,* mainly *Methylococcaceae* (*γ-Proteobacteria*) [58,59]. The latter increased ~10-fold in surface sediments of oxic treatments. Similarly, aerobic ammonium and nitrite oxidizers, such as *Nitrosomonadales* (*β-Proteobacteria*) [60] and *Nitrospirae* [61], increased by factors of ~7 and 2, respectively, in oxidized layers. In addition, sequence contributions of metabolically versatile *Burkholderiales* (*β-Proteobacteria*, mainly *Comamonadaceae)* [62,63] doubled, including sequences of the denitrifying genus *Pelomonas* [64], which suggests that besides nitrification also denitrification was enhanced in surface sediments of oxic cores. The increased importance of denitrification is matched by the increase in facultatively denitrifying $H_2S$ oxidizing *Thiotrichaceae* (*Thioploca, Beggioatoa*) and *Helicobacteraceae* (*Sulfuricurvum, Sulfurimonas*) [65] in surface sediments of oxic treatments.

Matching the geochemical impact of $O_2$, the increase in aerobic and denitrifying taxa in surface sediments of oxic treatments was accompanied by decreases in likely anaerobes (Figure 8). These include groups typically growing under anoxic or low $O_2$ conditions such as *Bacteroidia* [66] (*Bacteroidetes*), *Omnitrophica* [67], *Anaerolineae* [68] and *Dehalococcoidia* [69] (both *Chloroflexi*), *Clostridia* [70] (*Firmicutes*), and *Latescibacteria* [71,72]. In addition, $SO_4^{2-}$ reducing and syntrophic secondary fermenters such as *Syntrophaceae* (*Syntrophus, Desulfomonile, Smithella* and *Desulfobacca*) and $SO_4^{2-}$ reducing *Desulfarculaceae* (both *δ-Proteobacteria*) decreased [73,74]. The decrease in $SO_4^{2-}$ reducing taxa despite the increase in $SO_4^{2-}$ concentrations in oxic treatments can be explained with the anaerobic growth requirements of most sulfate reducers.

Despite the importance of microbial iron cycling in sediments of Lake Zurich [49], microbes involved in Fe cycling were generally low in relative abundances. Slight increases in ferrous Fe oxidizers (*Gallionellaceae, β-Proteobacteria, Nitrosomonadales*) [75] suggest an increase in aerobic iron oxidation in oxic treatments. Yet, other iron-oxidizing *Acidimicrobiales* (*Actinobacteria*) [76] decreased, and taxa that are known to include iron reducers, i.e., *Rhodobacteraceae* (*α-Proteobacteria*), *Holophagaceaa* (*Acidobacteria*), *Bacillaceae* (*Firmicutes*), *Deferribacteres* and *Geobacteraceae* (*δ-Proteobacteria,*), *Pseudomonaceae* and *Acidiferrobacteraceae* (both *γ-Proteobacteria*) [77] were rare (<0.5% mra) and showed no trend with treatments or sediment depth.

Although archaeal populations decreased in response to $O_2$, in line with a previous study reporting clear dominance of Bacteria over Archaea in $O_2$-exposed surface sediments [78], we observe no clear changes in archaeal community structure over time (Figures 8 and 9). A potential explanation is that changes in community composition to environmental changes, e.g., $O_2$ exposure, take longer in Archaea than Bacteria due to the long generation times of many archaeal taxa.

### 4.3. Effect of Chironomid Larval and Worm Bioturbation on Sediment Microbial Communities

Despite being comparatively minor, additional changes in microbial community composition occur in bioturbated treatments relative to macrofauna-free oxic treatments. Matching the more gradual redox profiles in L and W cores (Figures 2 and 4) and previous studies reporting correlations between redox potential and microbial community structure [35,79,80], bacterial communities in bioturbated cores change more gradually from oxidized surface sediment to reduced deeper layers compared to non-bioturbated treatments (Figure 9). By contrast, clear changes in nitrifying and denitrifying Bacteria (*Rhodocyclales, Nitrosomonadales, Burkholderiales* or *Rhizobiales*) or nitrifying Thaumarchaeota, whose activities have been previously reported to increase in bioturbated sediments [20,81–84], are absent. The only notable exception is perhaps the increased percentage of methane-oxidizing *Methylococcales* in surface sediment (0–0.5 cm) of W treatments (16%) compared to oxic non-bioturbated treatment (7%), but this was only observed at the final time point. Thus—under the experimental conditions studied—$O_2$ presence in bottom water had a

much greater impact on microbial community structure of bulk sediment than chironomid larval or tubificid worm bioturbation activity.

Instead, differences in microbial communities in bioturbated treatments were highly localized, being most evident in macrofauna themselves followed by macrofaunal feces and burrow structures (Figures 8 and 9). Consistent with a recent study on the same locations [37], faunal microbiomes were dominated by putatively parasitic as well as mutualistic $\alpha$-, $\beta$-Proteobacteria and/or Fusobacteria (worms), and $\alpha$-, $\beta$- and/or $\gamma$-Proteobacteria or Firmicutes (larvae) (for a detailed discussion of potential metabolic functions and host–microorganism interactions see [37]). Interestingly, microbial communities in larval tubes and burrows were very similar to those in oxidized surface sediments (Figures 8 and 9). This contrasts with previous studies documenting major differences in microbial communities between larval tubes and surrounding sediment [9,82] due to increased growth of methane-cycling microorganisms in tubes [9,15,85]. Rather than larval tubes or burrows, our study suggests that tubificid burrow communities, which have to our knowledge not been studied before, are clearly different from surrounding sediment. Compared to oxidized surface sediment or larval structures, worm burrows were more clearly dominated by $\beta$- and $\gamma$-Proteobacteria and also had higher percentages of $\varepsilon$-Proteobacteria. These differences in $\beta$-Proteobacteria were mainly due to an increase of Burkholderiales (unclassified Comamonadaceae and Oxalobacteraceae) and Methylophilales (unclassified Methylophilaceae) by a factor of 2–3. However most pronounced was the increase in $\varepsilon$-Proteobacteria, driven by a 16-fold increase in Sulfuricurvum (Helicobacteraceae) in worm burrows compared to oxic surface sediment. This suggests elevated proportions of aerobic and/or denitrifying, methylotrophic, and in particular S-oxidizing microbial taxa in worm burrows compared to surface sediment.

Microbial communities in worm and larval feces were very similar to those in corresponding burrow structures (Figures 8 and 9). The only conspicuous difference was an on average higher percentage of $\varepsilon$-Proteobacteria in worm feces. This increase was mainly driven by a ~4-fold increase of S-oxidizing Campylobacterales (Helicobacteraceae, Sulfuricurvum) compared to worm burrows. Given the proposed preferential metabolism of proteinaceous organic matter by tubificid worms, the relative increase in S-oxidizing microbial taxa in feces could be related to the elevated release of reduced inorganic S species by microbial fermentation of S-containing amino acids.

### 4.4. Possible Reasons for Weak Bioturbation Effects

The observed macrofaunal impact on microbial community structure was low but matched the overall minor biogeochemical impact of W and L compared to O treatments. Other studies have reported more pronounced effects of bioturbation on nutrient fluxes and microbial reaction rates in freshwater sediment [3,78,82].

A possible reason for the minor impact of chironomid bioturbation is that we used diverse, natural assemblies of chironomid species at close to natural abundances [37]. Chironomid larval taxa at the sampling site [37] are dominated by generalistic surface detritus-feeding and filter feeding taxa (e.g., Micropsectra sp., Tanytarsus sp.) [86], as well as predators (e.g., Procladius sp.). By comparison, most previous studies added mono-species communities of, e.g., Chironomus plumosus or C. riparius, at much higher population densities (e.g., 14,000 individuals per $m^2$ in [87] compared to ~1500 individuals $m^{-2}$ in our experiments), and with larger body size and body mass. The average body length of our chironomid larvae was around 1–1.5 cm, which is considerably smaller compared to 2–2.5 cm body lengths reported for 4th instar larvae of C. plumosus in many previous studies [50,88,89]. This size difference could be important, as bioirrigation rates are known to increase with size in chironomids [90].

As for L treatments, we also used natural assemblages of oligochaetes in W treatments. The dominant oligochaetes at the sampling site were L. hoffmeisteri and unidentified Tubificidae [37]. L. hoffmeisteri feeds mainly at 2–6 cm sediment depth and has been reported to have high reworking rates [91]. While we observed the typical head-down deposit feeding of tubificids in our experiments, we also observed the formation of galleries of worm

burrows to >20 cm sediment depth. Worms were frequently seen moving through these established burrow networks. Previous research has shown that these burrow networks are widespread in eutrophic lakes of central Switzerland and that sediment reworking by this mode of living is minimal [37]. Thus, the worm specimens that were used in the experiments could have engaged in unknown, deep feeding activities that only involved minimal sediment mixing (also see [37]). An additional factor may have been worm size. Previously it was shown that reworking activity and mixing depths are dependent on worm body length, and increase by an order of magnitude in worms with body lengths around 10 cm compared to worms with lengths around 5 cm [91,92]. Oligochaetes used here were typically ≤3 cm long, consistent with the observed low reworking rates. Furthermore, several, though not all, previous studies that have reported strong reworking by tubificids applied worm densities that were higher (e.g., 10,000 individuals $m^{-2}$ [93]) than in our experiments (~8500 individuals $m^{-2}$).

Finally, the impact of macrofaunal bioturbation on sediment geochemical conditions (especially solid phases) and microbial community structure might take longer than our study period of 3 months. Indeed, a recent study found long-term effects of chironomid bioturbation on P immobilization in sediment in a 3.5-yr experiment [18]. In our case, changes in bioavailable Fe pools were evident in W treatments during the second half of the experiments, indicating that mineral transformations due to bioturbation were not completed at the end of our experiments. Microbial generation times also need to be considered in the context of experimental duration, since these can vary greatly from hours and days [94] to weeks [95] to months or even longer in Archaea [96]. Slow population response times due to low growth rates could explain the rather slow change in microbial community structure, especially in archaeal communities. However, we did observe significant changes in microbial communities due to oxygenation and in macrofaunal burrows. Thus, slow microbial growth rates are unlikely to be the only reason for the overall minor impact of bioturbation on microbial communities in the sediments studied.

## 5. Conclusions

Our study provides novel insights into the impacts of reoxygenation and macrofaunal recolonization on sediment biogeochemistry and microbial community composition. Based on mesocosm experiments involving intact sediment cores from a previously hypoxic and macrofauna-free lake basin, we show that oxygenation of bottom water alone has a stronger impact on sediment biogeochemistry and microbial community structure than the additional presence of bioturbating macrofauna. The comparatively smaller impact of bioturbation in the sediments studied than observed in numerous past studies could be in part due to the use of natural chironomid larval and tubificid worm communities at close-to-natural densities and size distributions in our work. By contrast, most previous studies on this topic have involved sediment amendments with mono-species assemblages of larvae or worms, at population densities and size distributions that are considerably higher than those in natural sediments of the lakes studied. Though more research, e.g., involving longer time scales, more locations, and additional experimental setups is necessary, our study demonstrates the feasibility and importance of using natural communities and population densities to study their impact on elemental cycles and their microbial agents in lake sediments.

**Supplementary Materials:** The following are available online at https://www.mdpi.com/article/10.3390/w13111599/s1. Supplementary Materials (SM) Text S1: Redox state of sediment (EAC and EDC), Text S2: Nucleic acid extraction from sediment, Text S3: Methodology: reverse transcription and quantification of rRNA gene copies, Text S4: Quantitative polymerase chain reaction (qPCR), Text S5: Next Generation Sequencing (NGS), Text S6: Single burrow oxygen profiles, Text S7: Microbial community zonation across treatments; Figure S1: pictures of the sediment cores in the tank during the experiment. A: side view, B: top view. Sediment cores with lids are the hypoxic controls. Figure S2: pH microsensor profiles during the course of the experiment, all profiles recorded on monitoring and sampling days are shown, cores to be sampled next are marked in black. Figure S3: $H_2S$ (lower x-axis) and $O_2$ (upper x-axis) microsensor profiles at T6 along sediment depth in microm-

eter ($\mu$m). In each core five profiles for $O_2$ and $H_2S$ were measured except in the hypoxic controls where only one profile each was obtained. Please note for the oxic treatment two separate cores were measured (10 profiles of $O_2$ and $H_2S$). Please also note $H_2S$ concentrations reflect $H_2S$ and not total sulfide concentrations, as they were not corrected with pH. Figure S4: $O_2$ microsensor profiles during the course of the experiment, all profiles recorded on monitoring and sampling days are shown, cores to be sampled next are marked in black. Figure S5: representative single burrow $O_2$ gradients measured in L (upper panel) and W (lower panel) treatments at different points in the core (profiles a-c), measured at T2. Figure S6: Redox potential (Eh) profiles during the experiment. Cores to be samples next are marked in black, whereas monitoring profiles are marked in grey lines. Figure S7: Porewater profiles of all measured anions and cations at all timepoints. Please note concentration ranges on x-axis differ between different analytes. Figure S8: Concentration profiles of $NH_4^+$. The sediment–water interface is indicated with a horizontal line. Figure S9: Bioavailable Fe(II) and Fe(III) concentrations in $\mu$mol $g^{-1}_{ww}$ in the upper 4 cm of the sediment, extractions were performed with 0.5 M HCl. Figure S10: EAC and EDC measurements from mediated electrochemical analysis (MER and MEO) in number (nb) electrons (e-). Figure S11: Luminophore bead counts at T1 (6 d), T3 (27 d), T5 (63 d), and T6 (82 d). Displayed as% of total (sum of the whole profile) for each treatment. Subplots show zoom in on the upper 2 cm of the core. Figure S12: Chlorophyll a (Chla) (A) and Pheopigments (Pheo) (B) in $\mu$g gww$^{-1}$ as well as the ratio of Chla to Pheo (C) at T6 (82 d) for all treatments vs. sediment depth (cmblf). Figure S13: Left: bacterial ($DNA_{bac}$), and right: archaeal ($DNA_{arc}$) copy numbers and Bacteria to Archaea ratios (BARs) plotted against sediment depth. Left: BAR calculated from DNA, right: BAR calculated from cDNA. Each row corresponds to one different time-point (T3 (27 d), T5 (63 d) and T6 (82 d)) with t0 as common reference line (same data), Figure S14: Relative abundances of Bacteria based on RNA extracts on the phylum level vs. sediment depth (cmblf). Columns show different treatments (C, O, W, L), rows show timepoints (T3 (27 d), T5 (63 d), T6 (82 d)). Figure S15: Relative abundances of Archaea based on RNA extracts on the phylum level vs. sediment depth (cmblf). Columns show different treatments (C, O, W, L), rows show timepoints (T3 (27 d), T5 (63 d), T6 (82 d)). Figure S16. Left: CAP plots on bacterial phylum-level based on weighted-UniFrac: (A) samples from all time points and all treatments, (B) T6 samples from all treatments. Right: CAP plots on archaeal phylum-level based on weighted-UniFrac: (A) samples from all time points and all treatments, (B) T6 samples from all treatments; Table S1: Bacteria to Archaea ratios (BARs) calculated from qPCR and from reads obtained from NGS with standard deviations and bacterial and archaeal read numbers according to sample type. Number of values for each category can be found in parenthesis. Table S2: Table of relative abundance on the Phylum, Class and Order level for Archaea and Bacteria, with mean relative abundance (mra), maximum (Max) and the overall maximum (Overall), all T6 (82 d).

**Author Contributions:** Conceptualization, A.F. and M.A.L.; data curation, A.F., A.G., S.G. and M.A.L.; formal analysis, A.F., A.G., S.G., D.M., P.J. and M.S.; funding acquisition, M.A.L.; investigation, A.F., A.G., S.G., D.M. and P.J.; methodology, A.F., A.G., S.G., D.M., P.J., M.S., C.J.S. and M.A.L.; project administration, M.A.L.; resources, C.J.S. and M.A.L.; validation, A.F., A.G., S.G., D.M. and M.A.L.; visualization, A.F., A.G., S.G. and D.M.; writing—original draft, A.F., A.G., S.G. and M.A.L.; writing—review and editing, A.F., A.G., S.G., D.M., P.J., M.S., C.J.S. and M.A.L. All authors have read and agreed to the published version of the manuscript.

**Funding:** This research was funded by Swiss National Science Foundation, grant number 205321_163371 to M.A.L.

**Institutional Review Board Statement:** Not applicable.

**Informed Consent Statement:** Not applicable.

**Data Availability Statement:** The data presented in this study are available on request from the corresponding authors.

**Acknowledgments:** We thank Laurel Kathleen Thomas Arrigo, Christoph Höfer and Ruben Kretzschmar for helpful advice during experiments and analysis. Special thanks to Alois Zwyssig, Beat Kienholz and all technicians from EAWAG (Kastanienbaum, CH) for great help with core and macrofauna sampling and the experimental set up. Many thanks to Iso Christl and Jasmine Berg for consultations during measurements and analyses. We further want to thank the Genetic Diversity Center (GDC) at ETH Zurich for help with library preparation and next generation sequencing.

**Conflicts of Interest:** The authors declare no conflict of interest.

## Appendix A

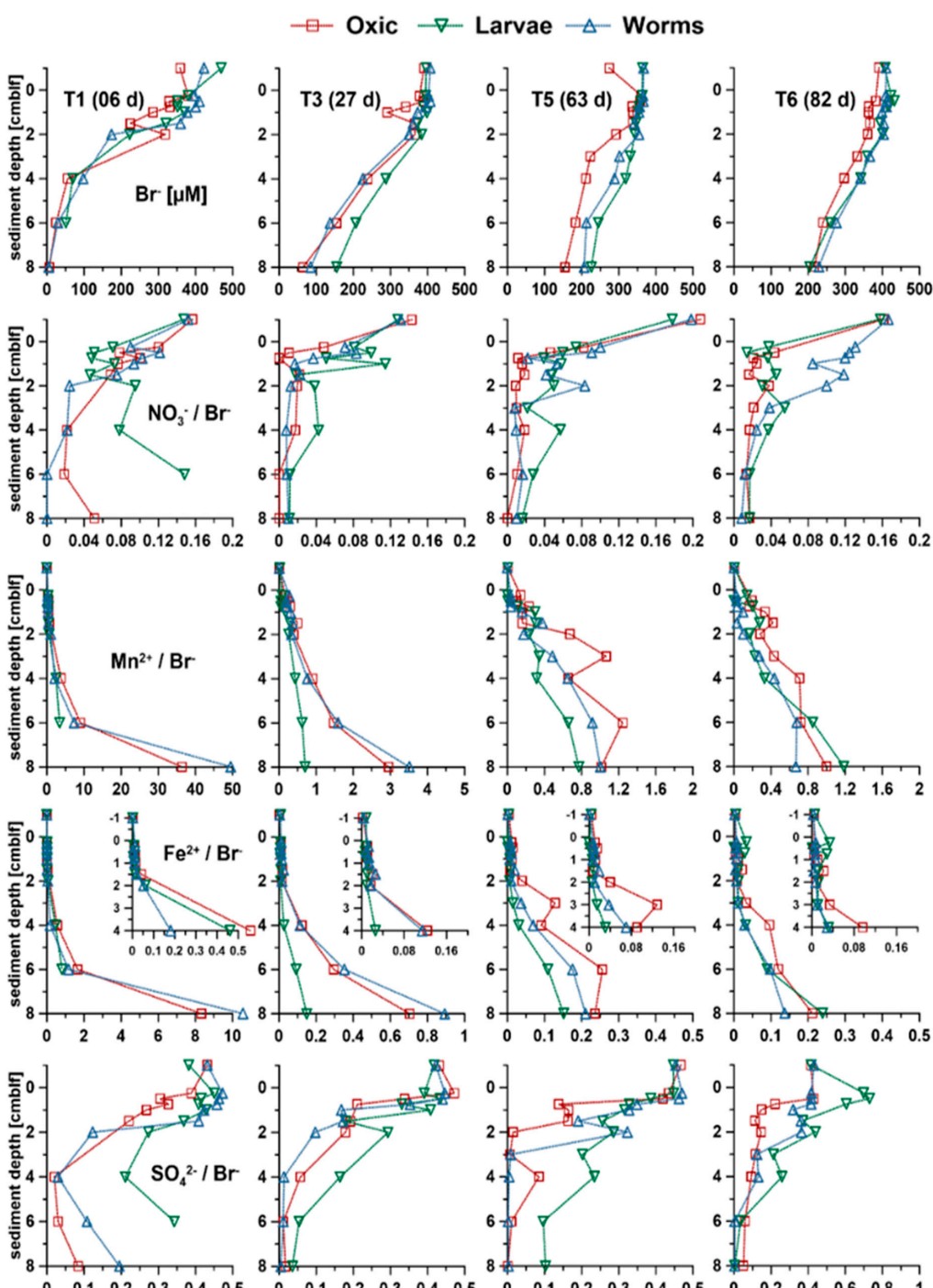

**Figure A1.** Ratios of pore water species/Br$^-$, please note due to ratio increases and decreases over time, x-axes ranges change.

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
