# Peer review of "Effects of Macrofaunal Recolonization on Biogeochemical Processes and Microbiota—A Mesocosm Study"

_water, doi:10.3390/w13111599_

Round 1
Reviewer 1 Report
General Comments This is an excellent paper. It is well-written, informative, and interesting. The methods are clearly written, and the conclusions support the results. I applaud the use of community data rather than population data in this study even if the authors were unable to show a strong influence of the chironomids and oligochaetes in driving microbial communities and lake sediment biogeochemistry. This is a strength of the paper. My only comments relate to some formatting issues which may be from the journal or the original manuscript. Overall, some references to figures are noted as errors. Some references to figures are in bold, whereas others are not in bold. The sub-headers are not all aligned nor are they clearly differentiated from the following text in some places. My recommendation is to thoroughly review the formatting to fix these issues. I provide a brief list of some problems identified while reviewing the manuscript. Specific comments: Line 202 Sub-header needs to drop down to next page. Line 222 Sub-header does not line up with sub-header on line 202. Line 253 Space issue between 2.5 and ml Line 274 Sub-header alignment Lines 304, 305, 310-313, 481, 490, 544, 548, 552, 557, 559, others Error message referring to figure or table. Line 539 Sub-header formatting?
Reviewer 2 Report
The paper focused on effects of macrofaunal recolonization on biogeochemical processes and microbiota investigated in elegant microcosm experiment provides a new insight into ecological mechanism of interaction between benthic macrofauna and bacteria depending on dissolved oxygen availability.
The research is appropriately designed, methodologically complex, results are clearly presented and manuscript well written. Maybe it is a problem of the system, but in my pdf version there are plenty of notes: “Error! Reference source not found” and in fact I could not find reference to the figures. Statistics for geochemical analyses are highly recommended.
I recommend to publish the paper in Water after minor revision.
Below I present a few suggestions.
P3, l 103-112 I suggest to move this part of the text into the method section
P3, l 112-115 I suggest to move this part of the text into the discussion section
P3, l 135 please introduce “Chironomid larvae treatment (L)”
P3, l 139 I suggest to delete time labels (T1 – T6) consequently in whole text and appendix. Days are more clear to readers.
P5, l 185-191 please add description of letters marking treatments
P6, l 256 the abbreviation is used for the first time, please add a full meaning
P8, l 321 I suggest to change the title “oxygen concentration and redox potential”
P10, l 364-366, 382-383 these results should be mentioned in the abstract
P17, l 539 use italics as it is the subtitle
P18, l 603 add “respectively”
P19, l 663-665 please use more clear sentence
Figure A4 and A6 please rotate the figures, so the axis title will be in the proper position
